

# PHFinder: assisted detection of point heteroplasmy in Sanger sequencing chromatograms

Marcos Suárez Menéndez[1], Vania E. Rivera-León[1], Jooke Robbins[2], Martine Berube[1,2] and Per J. Palsbøll[1,2]

[1] Marine Evolution and Conservation, Groningen Institute for Evolutionary Life Sciences, University of Groningen, Groningen, The Netherlands
[2] Center for Coastal Studies, Provincetown, MA, United States of America

## ABSTRACT

Heteroplasmy is the presence of two or more organellar genomes (mitochondrial or plastid DNA) in an organism, tissue, cell or organelle. Heteroplasmy can be detected by visual inspection of Sanger sequencing chromatograms, where it appears as multiple peaks of fluorescence at a single nucleotide position. Visual inspection of chromatograms is both consuming and highly subjective, as heteroplasmy is difficult to differentiate from background noise. Few software solutions are available to automate the detection of point heteroplasmies, and those that are available are typically proprietary, lack customization or are unsuitable for automated heteroplasmy assessment in large datasets.

Here, we present PHFinder, a Python-based, open-source tool to assist in the detection of point heteroplasmies in large numbers of Sanger chromatograms. PHFinder automatically identifies point heteroplasmies directly from the chromatogram trace data. The program was tested with Sanger sequencing data from 100 humpback whales (*Megaptera novaeangliae*) tissue samples with known heteroplasmies.

PHFinder detected most (90%) of the known heteroplasmies thereby greatly reducing the amount of visual inspection required. PHFinder is flexible and enables explicit specification of key parameters to infer double peaks (*i.e.*, heteroplasmies).

## INTRODUCTION

Heteroplasmy is the presence of multiple organellar (mitochondrial or plastid) genomes in an organism, tissue or cell. Despite advances in so-called next-generation sequencing, Sanger sequencing (*Sanger, Nicklen & Coulson, 1977*) is still widely employed in studies targeting specific highly-variable organellar DNA regions, such as the mitochondrial control region. The DNA sequence is inferred from the resulting chromatogram, where the base at each nucleotide position is represented by a fluorescent signal of base-specific colour (each representing a different deoxynucleotide). Heteroplasmy, due to point mutations, is apparent as two fluorescent peaks in the same nucleotide position. All other factors being equal, the relative height of each fluorescent peak reflects the relative abundance of each deoxynucleotide, and, by extension, the two mtDNA haplotypes (*Irwin et al., 2009*).

Corresponding author
Marcos Suárez Menéndez,
m.suarez.menendez@rug.nl

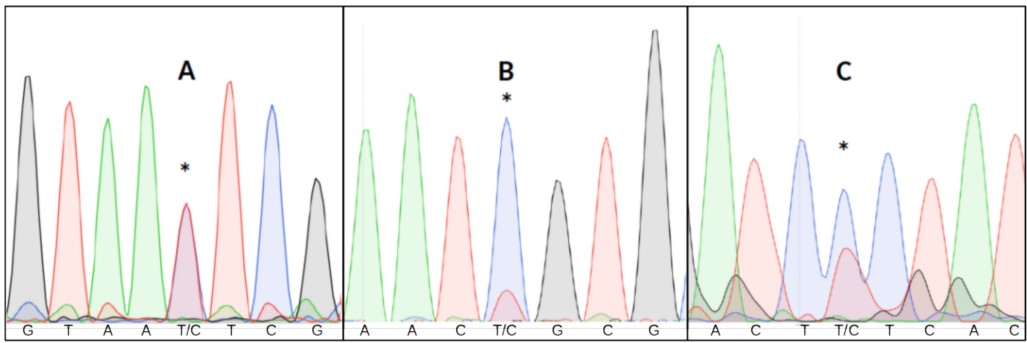

**Figure 1** **Three examples of point heteroplasmy in different chromatograms.** An asterisk (*) indicates Putative heteroplasmies. (A) Completely overlapping fluorescent peaks (likely heteroplasmy). (B) Significantly lower secondary fluorescent peak (likely background noise). (C) Heteroplasmy with background noise, making it more difficult to detect.

Heteroplasmy due to insertions or deletions results in multiple fluorescent peaks at several consecutive nucleotide positions (length heteroplasmy).

Detecting heteroplasmy is necessary when studying certain mitochondrial diseases (*Stewart & Chinnery, 2015*), conducting forensic work (*Salas, Lareu & Carracedo, 2001*) or estimating mitochondrial mutation rates (*Millar et al., 2008*). Failing to take heteroplasmy into consideration can also introduce errors in other kinds of studies. For example, in relatedness studies, maternal relatives might appear to have different mitochondrial haplotypes due to different heteroplasmic proportions (*Klütsch et al., 2011*). Visual inspection of all chromatograms to detect putative double fluorescent peaks is impractical in large datasets, and difficult to replicate given the subjective nature of the assessment (Fig. 1).

Previous studies have applied different criteria to infer heteroplasmies from the ratio of the two fluorescent peaks in a putative heteroplasmic nucleotide position (*e.g.*, >10%, *Brandstätter, Niederstätter & Parson, 2004*; *Irwin et al., 2009*; or >30%, *Baker et al., 2013*) or altogether omitted (*e.g.*, *Vollmer et al., 2011*).

Existing software can facilitate the automatic detection of double fluorescent peaks in chromatograms. These are either proprietary (*e.g.*, SEQUENCHER, GeneCodes Inc., Ann Arbor, MI) or lack customization and tend to disregard some double fluorescent peaks as background noise, or are unable to process large datasets (*e.g.*, SNAPGENE® VIEWER v4.3.7, GSL Biotech LLC, Boston, MA, USA). We developed a bioinformatic pipeline (point heteroplasmy finder, PHFinder) as a means to screen DNA chromatograms in the commonly employed AB1 format (*Applied Biosystems Inc, 2006*); generated by DNA sequencers, such as the Applied BiosystemsTM Genetic Analyzer series (Thermo Fisher Scientific Inc., Waltham, MA, USA) to detect double fluorescent peaks in an automated manner. PHFinder facilitates the detection of point heteroplasmies by applying filters using average base call quality scores and the level of background noise in a user-specified target region of the relevant DNA sequence. Portions of this text were previously published as part of a preprint (https://www.biorxiv.org/content/10.1101/2022.08.17.501710v1).

## Implementation

PHFinder was written in Python v3.6.8 (*Van Rossum & Drake Jr, 1995*) and BASH (*Ramey & Fox, 2016*). PHFinder dependencies include Biopython v1.73 (*Cock et al., 2009*) and BOWTIE2 (*Langmead & Salzberg, 2012*).

First, a FASTQ file (sequence of nucleotides and the associated Phred quality scores *Ewing et al., 1998*) is extracted from each AB1 file. FASTQ files are subsequently aligned against a reference sequence with BOWTIE2 and the result is saved in a single Sequence Alignment Map (SAM) format file (*Li et al., 2009*) per alignment. The orientation of the chromatogram (forward or reverse) and starting point of the reference sequence is subsequently extracted from each SAM file in order to position the trace information to the correct region in each chromatogram.

The data elements stored with AB1 files are associated with specific tags. PHFinder uses the information contained in the AB1 tags; DATA9 to DATA12 (trace information for guanine, adenine, thymine and cytosine); PBAS2 (the sequence of base calls); PLOC2 (location of base calls) and PCON2 (per-base call quality score) as specified in the original file fomat (*Applied Biosystems Inc, 2006*).

The presence of a double fluorescent peak (*i.e.*, a potential heteroplasmy) was inferred from the values of three *ad hoc* indexes calculated from the above data:

1. Average base call quality (AQ) of the bases in the targeted DNA sequence region (measured as Phred quality scores, ranging from 0 to 93).
2. Main ratio (MR) of a double fluorescent peak, *i.e.*, the height of the second highest peak as a fraction of the highest peak (Fig. 2).
3. Secondary ratios (SR) of the three down- and upstream nucleotide positions flanking the putative heteroplasmic position; estimated as the height of the second highest peak as a fraction of the highest peak (Fig. 2).

A position is inferred to be heteroplasmic by PHFinder if the three indexes described above exceeds user determined threshold values (Fig. 2).

## MATERIAL AND METHODS

A test set of mitochondrial control region DNA sequences were determined in DNA extracted from 100 skin samples collected during a long-term study of individual humpback whales (*Megaptera novaeangliae)* in the Gulf of Maine (North Atlantic). DNA sequence data were randomly selected from 30 samples with predetermined heteroplasmies as well as from 70 samples that appeared homoplasmic. Heteroplasmies were identified based on comparison to samples from close maternal relatives (known through longitudinal studies of individuals or microsatellite markers) or experimental confirmation using the dCAPS technique (*Neff et al., 1998*, data not shown).

Skin samples were collected by biopsy techniques (*Palsbøll, Larsen & Hansen, 1991*), under U.S. NOAA, ESA/MMPA permits 787, 633-1483 and 633-1778, and stored in 5 M NaCl with 25% DMSO (dimethyl sulfoxide, *Amos & Hoelzel, 1991*) at -20/-80 degrees Celsius (°C) prior to DNA extraction. Total-cell DNA was extracted by standard phenol/chloroform extractions as described by *Russel & Sambrook (2001)* or using QIAGEN

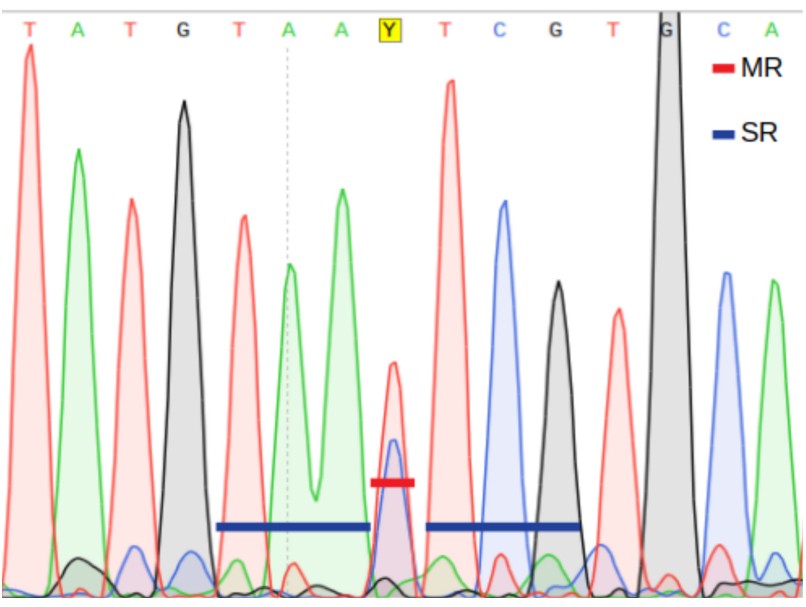

**Figure 2** Main ratio (MR) and secondary ratio (SR) indexes.

DNEasyTM Blood and Tissue Kit (QIAGEN Inc., Hilden, Germany) following the manufacturer's instructions. Extracted DNA was stored in 1xTE buffer (10 mM Tris–HCl, 1mM EDTA, pH 8.0) at −20 °C.

The sequence of the first 500 base pairs (bps) of the 5′ end of the mitochondrial control region was determined as described previously by *Palsbøll et al. (1995)* using the oligo-nucleotide primers BP16071R (*Drouot et al., 2004*) and MT4F (*Arnason, Gullberg & Widegren, 1993*). Unincorporated nucleotides and primers were removed from the polymerase chain reactions (PCR, *Mullis & Faloona, 1987*) with Shrimp Alkaline Phosphatase and Exonuclease I, as described by *Werle et al. (1994)*. Subsequent cycle sequencing conducted with the above-mentioned nucleotide primers and the BigDye® Terminator v3.1 Cycle Sequencing kit (Applied Biosystems Inc., Waltham, MA, USA) following the manufacturer's protocol. The cycle sequencing products were precipitated with by ethanol and sodium (*Russel & Sambrook, 2001*). The order of sequencing fragments was resolved by electrophoresis on an ABI 3730 DNA Analyzer® or and ABI PRISM® 377 DNA Sequencer (Applied Biosystems Inc., Waltham, MA, USA).

All chromatograms were visually inspected for point heteroplasmies using SNAPGENE® VIEWER (v4.3.7, GSL Biotech LLC). PHFinder was tested and validated (on GNU/Linux systems) by analysing the dataset with different threshold values for each index (MR: 15, 25, 35, 45; SR: 0.2, 0.3, 0.4, 0.5; and AQ: 30, 40, 50, 60) resulting in 64 different combinations of index threshold values. The results of each index threshold combination were visualized to assess which combination was most efficient in detecting the known heteroplasmies; specifically, the number of detected heteroplasmic AB1 files versus the number of false positives (nucleotide positions that were incorrectly deemed as heteroplasmic).

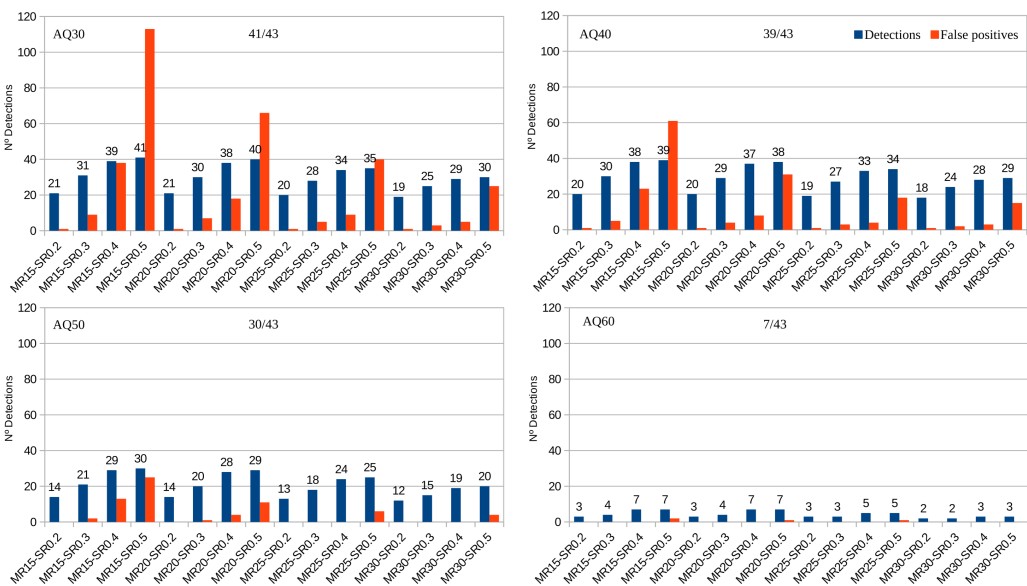

**Figure 3** Number of detected heteroplasmic AB1 files from a total of 43 *vs.* false positives for each combination of index threshold values divided according to AQ indexes (30, 40, 50, 60).

## RESULTS

The data from the 100 samples comprised 189 AB1 files (tissue samples were sequenced between 1-6 times). The first 500 bps of the mitochondrial control region of *M. novaeangliae*, started at position 15,490 and ended at position 15,970 according to the reference mitochondrial genome sequence NC_006927.1 published by *Sasaki et al. (2005)*, which was also used as reference in the alignment. The analyses conducted here targeted the region from position 15,540 to position 15,815 (275 bps).

Among the 188 AB1 files, PHFinder was unable to process five (due to software incompatibilities with data generated by older DNA sequencers) and another 19 files were empty. Out of the 100 samples, 30 samples (43 AB1 files) contained known point heteroplasmies at seven different nucleotide positions with MRs ranging from 18 to 88 (Table S1). Figure 3 shows the fraction of these known heteroplasmies detected by PHFinder for each combination of index threshold values as well as the number of false positives.

The number of samples and AB1 files included in each analysis varied with the AQ index: AQ 30, 100 samples and 143 AB1 files. AQ 40, 88 samples and 115 AB1 files. AQ50, 68 samples and 79 AB1 files. AQ60, 37 samples and 15 AB1 files. Detailed results for each combination of index threshold values are shown in Table S2.

## DISCUSSION

PHFinder was developed to assist the detection of point heteroplasmies in large data sets. The program automates a first pass of the data, reducing the number of AB1 files that need to be visually inspected. Although PHFinder detected most of the point heteroplasmies present in the dataset (95.3%), the present analysis revealed some limitations.

The samples were randomly selected from a large dataset in order to include a wide range of DNA sequences in terms of overall DNA sequence quality, length, DNA strand sequenced, as well as corrupted or older AB1 files and different MR values in order to test PHFinder under realistic conditions. The samples were represented by different numbers of AB1 files as some samples were re-sequenced several times (*i.e.*, because of bad quality or to sequence both DNA strands). Among the 100 samples, 30 samples contained a known heteroplasmic nucleotide position to ensure sufficient data for testing PHFinder.

PHFinder was unable to process five AB1 files due to Biopython's v1.73 (*Cock et al., 2009*) inability to access AB1 files generated by older DNA sequencers (ABI PRISM$^{®}$ 377 DNA Sequencer in this instance). Potential compatibility issues could be due to differences in the tags in AB1 files, and hence resolved by modifying the tag names in the PHFinder main script.

The PHFinder assessment targeted the region from position 15,540 to position 15,815 (275 bps) in order to avoid regions of chromatograms with elevated background noise (*i.e.*, the 5′ and 3′ ends). This strategy reduced the proportion of false positives of heteroplasmies as the targeted region usually presents higher average qualities.

Unsurprisingly, the main limiting factor was the average base call quality of the AB1 files. Low average base call quality is mainly due to elevated background noise which, in turn, may be erroneously inferred as putative heteroplasmies. These kinds of false positives were easily recognised as artefacts during subsequent visual inspection of the chromatograms. Since all putative heteroplasmies highlighted by PHFinder should be visually confirmed, the most efficient approach is to employ a combination of index threshold values that yields the highest number of heteroplasmies and lowest number of false positives (for these data; MR: 20, SR: 0.4 and AQ: 40, Fig. 3). In this study the aforementioned combination of index threshold values identified 27 (out of 30 known heteroplasmies in 37 AB1 files) and only eight false positives.

Employing low AQ index threshold values increased the number of AB1 files (and the corresponding samples) in an assessment, without a similar increase in heteroplasmy detection, *i.e.*, the overall frequency of heteroplasmy detections was reduced (*e.g.*, Fig. 3, AQ30 vs. AQ40). We observed a clear trade-off between the number of detected heteroplasmies and false positives. Low PHFinder index threshold values (*i.e.*, low MR, SR, and AQ) increased the number of detected heteroplasmies but also the number of false positives. Since all samples with putative heteroplasmies require visual inspection, lowering the threshold index values led to increasing amounts of visual inspection. Applying higher index threshold values (*i.e.*, a high MR, SR and AQ) had the opposite effect, *i.e.*, fewer false positives and less visual inspection accompanied with a lower detection rate of heteroplasmies.

The SR filter is aimed at filtering out positions where an heteroplasmic double peak can not be distinguish from the surrounding background noise in a chromatogram. This filter is useful when there is background noise in the chromatograms but can exclude the detection of multiple heteroplasmies if they are closer than three base pairs. The filter can be effectively deactivated by setting a high threshold (*e.g.*, 100) but with the trade-off of

potentially introducing more noise in the results if the chromatograms do not have very high quality.

Heteroplasmies with a MR as low as 15% could be potentially detected, based on the detection limit of Sanger sequencing (*Tsiatis et al., 2010*). Although the lowest MR in this study was estimated to 18%, (Table S1), the lowest MR index threshold applied was 15% in order to assess the effect on the rate of false positives.

The optimal index threshold value combination will likely depend upon the aim of a study and the overall quality of the DNA sequence chromatograms. If the goal is to compare heteroplasmy frequencies, then a fairly strict index threshold value combination can be employed to reduce the number of false positives. In more detailed assessments that are aimed at identifying as many heteroplasmies as possible (*e.g.*, to detect, novel deleterious mutations), a lower index threshold value combination will facilitate a higher heteroplasmy detection rate, but require elevated levels of post-analysis visual inspection. If the targets are specific, known heteroplasmies (*i.e.*, in a known nucleotide position), a lowered index threshold value combination can be employed as only putative heteroplasmies in the targeted positions would require post-analysis visual inspection.

Once double peaks are detected the potential causes need to be considered. Sequencing artefacts as well as contamination from other samples could cause double peaks in chromatograms. This is especially important when heteroplasmy has not previously reported in the study organism. Re-extracting the sample and/or sequencing both the forward and the reverse strand can help resolve such issues (*Rodríguez-Pena et al., 2020*). Biological factors can also result in double peaks, such as parental leakage (*Pearl, Welch & McCauley, 2009*), mutations (*Suárez-Menéndez et al., 2023*) and the presence of nuclear mitochondrial DNA segments (*Wallace et al., 1997*). The probability of these events varies among species and it may affect the pattern of heteroplasmy and the detection accuracy (*e.g.*, multiple heteroplasmic positions in close proximity of each other). It is key to be aware of the peculiarities inherent to the specific organism. In some cases, it may be necessary to further confirm the putative heteroplasmies whether by re-extraction, re-sequencing, next generation sequencing or dCAPS.

The main advantage of PHFinder is the ability to customize assessments on a case-by-case basis. Written in Python, the source code can easily be changed and improved to fit specific research needs. PHFinder provides the ability to implement and apply explicit assessment criteria (in terms of base quality and fluorescent peak ratios), thereby facilitating direct objective comparisons among different data sets. Although a final visual inspection of chromatograms with putative heteroplasmic sites will always be required, PHFinder greatly reduces the number of chromatograms requiring visual inspection which is especially valuable in large datasets.

## Data accessibility

PHFinder scripts and all data used in this article are available online from https://github.com/MSuarezMenendez/PHFinder and in Zenodo https://doi.org/10.5281/zenodo.8159009, as well as instructions of how to use PHFinder on the command line.

### Funding

Marcos Suárez Menéndez was supported by a doctorate fellowship from the University of Groningen. The funders had no role in study design, data collection and analysis, decision to publish, or preparation of the manuscript.

### Grant Disclosures

The following grant information was disclosed by the authors:
The University of Groningen.

### Competing Interests

The authors declare there are no competing interests.

### Author Contributions

- Marcos Suárez Menéndez conceived and designed the experiments, performed the experiments, analyzed the data, prepared figures and/or tables, authored or reviewed drafts of the article, and approved the final draft.
- Vania E. Rivera-León conceived and designed the experiments, authored or reviewed drafts of the article, and approved the final draft.
- Jooke Robbins performed the experiments, authored or reviewed drafts of the article, collected tissue samples and field data used in the study, and approved the final draft.
- Martine Berube conceived and designed the experiments, performed the experiments, authored or reviewed drafts of the article, and approved the final draft.
- Per J. Palsbøll conceived and designed the experiments, authored or reviewed drafts of the article, and approved the final draft.

### Animal Ethics

The following information was supplied relating to ethical approvals.
Biopsy samples collected under the U.S. NOAA ESA/MMPA permits (787, 633-1483, and 633-1778).

### DNA Deposition

The following information was supplied regarding the deposition of DNA sequences:
The raw chromatograms and FASTQ files extracted from them are available in the Supplemental File. The FASTQ files (that include the DNA sequence) for each chromatogram are generated as a by-product of the analysis but they are not a result used in any analysis.

## Data Availability

The raw data used for the analysis (AB1 files) are available in the Supplementary Files.

The PHFinder scripts, instructions of how to use PHFinder on the command line, and all data used in this article are available at GitHub and Zenodo:

- Available at https://github.com/MSuarezMenendez/PHFinder

- MSuarezMenendez. (2023). MSuarezMenendez/PHFinder: PHFinder v1.0.0 (v1.0.0). Zenodo. Available at https://doi.org/10.5281/zenodo.8159009.

## Supplemental Information

Supplemental information for this article can be found online at http://dx.doi.org/10.7717/peerj.16028#supplemental-information.

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
