# Peer review of "PHFinder: assisted detection of point heteroplasmy in Sanger sequencing chromatograms"

_PeerJ, doi:10.7717/peerj.16028_

## Round 0.1 · original submission · Minor Revisions

Dear Dr. Menéndez and colleagues:

Thanks for submitting your manuscript to PeerJ. I have now received two independent reviews of your work, and as you will see, the reviewers raised some concerns about the research. Despite this, these reviewers are optimistic about your work and the potential impact it will have on research studying methods for facilitating DNA sequence data analysis. Thus, I encourage you to revise your manuscript, accordingly, taking into account all of the concerns raised by both reviewers.

While the concerns of the reviewers are relatively minor, this is a major revision to ensure that the original reviewers have a chance to evaluate your responses to their concerns. There are many suggestions, which I am sure will greatly improve your manuscript once addressed.

Please note that reviewer 1 has included a marked-up version of your manuscript.

Please ensure to make all aspects of your research reproducible; specifically, please clarify the points raised by the issues that need more explanation. Also, ensure that data are submitted to your public repository.

Therefore, I am recommending that you revise your manuscript, accordingly, taking into account all of the issues raised by the reviewers.

Good luck with your revision,

-joe

Reviewer 1 ·

Basic reporting

No comment, article is well written and clear.

Experimental design

No comment, see "additional comments" for things that could be clarified.

Validity of the findings

No comment, this is a useful contribution to the field.

Additional comments

This is a useful contribution to DNA sequence analysis. Although I’m not sure how much it will speed up things for those who do take the time to carefully check their chromatograms, it will hopefully gain wider notice and use, as my experience is that many do not. It is clearly written and for the most part the results are well presented. I have only a few comments, listed below by the line number in the manuscript file, which I have uploaded as a pdf with comments (the line numbers in the pdf sent to review are slightly different so I have given those as well).

line 90 (pdf: 93) – The use of secondary ratios of the adjacent peaks seems like it can potentially be a problem if other peaks are heteroplasmic? This can be an issue with a third position polymorphism followed by a first position. In addition, there are not necessarily only two haplotypes.

line 100 (pdf: 104) – I am not familiar with dCAPS, and the other data is not shown, but my understanding of it is that it only confirms that heteroplasmies exist. Was this tested on the bases that were marked as false positives? See e.g. Magnacca & Brown 2010 (10.1111/j.1755-0998.2009.02724.x) for examples where samples that appear homoplasmic were in fact heteroplasmic.

line 136 (pdf: 140) – Give a brief mention why the five files could not be processed; maybe say “due to apparent software incompatibility” or something and refer to the longer explanation in the discussion.

line 139 (pdf: 143) – The figure shows the number of heteroplasmies, not the proportion. If I am reading Table S1 correctly, there are 43 total. It would be helpful to have Figure 3 show the total number available for each graph, since it varied between AQ tests. This would allow the reader to better gauge the accuracy of the methods, since it is the proportion found (and inaccurate) that is important. In addition, a more useful way to visualize it would be to make the Y axis 0–70 for the AQ30, 40, and 50 graphs, which would only require one over-long line (for MR15-SR0.5), and one graph with a different scale (AQ60, the one that clearly was not very good).

Annotated reviews are not available for download in order to protect the identity of reviewers who chose to remain anonymous.

Reviewer 2 ·

Basic reporting

The figures are clear, although there is an editing issue in Figure 1, particularly in example C, which indicates image manipulation. This aspect needs to be addressed.

Experimental design

No comment

Validity of the findings

The authors demonstrate a comprehensive understanding of the tool's limitations and emphasize that, although it reduces human efforts, visual inspection of the data is still necessary. However, it would be valuable for the authors to discuss the other potential origins of the double peaks, such as sequencing artifacts or contaminations, and emphasize the significance of taking these factors into account, especially in cases where heteroplasmy has not been previously documented for the sampled organism/dataset.

Additional comments

Although the raw data for testing is provided as supplementary material, it would be beneficial to also include them on the GitHub examples page (either the files themselves or a link to access the data).

---

## Round 0.2 · accepted · Accept

Dear Dr. Menéndez and colleagues:

Thanks for revising your manuscript based on the concerns raised by the reviewers. I now believe that your manuscript is suitable for publication. Congratulations! I look forward to seeing this work in print, and I anticipate it being an important resource for groups studying methods for facilitating DNA sequence data analysis. Thanks again for choosing PeerJ to publish such important work.

Best,

-joe